# Learning and Aligning Structured Random Feature Networks

**Vivian White**
Department of Computer Science
Western Washington University
Bellingham, WA, USA
`whitev4@wwu.edu`

**Muawiz Chaudhary**
Department of Computer Science
Mila, Concordia University
Montréal, QC, Canada

**Guy Wolf**
Department of Mathematics and Statistics
Mila, Université de Montréal
Montréal, QC, Canada

**Guillaume Lajoie**
Department of Mathematics and Statistics
Mila, Université de Montréal
Montreal, QC, Canada

**Kameron Decker Harris**
Department of Computer Science
Western Washington University
Bellingham, WA, USA
`kameron.harris@wwu.edu`

## Abstract

Artificial neural networks (ANNs) are considered "black boxes" due to the difficulty of interpreting their learned weights. While choosing the best features is not well understood, random feature networks (RFNs) and wavelet scattering ground some ANN learning mechanisms in function space with tractable mathematics. Meanwhile, the genetic code has evolved over millions of years, shaping the brain to develop variable neural circuits with reliable structure that resemble RFNs. We explore a similar approach, embedding neuro-inspired, wavelet-like weights into multilayer RFNs. These can outperform scattering and have kernels that describe their function space at large width. We build learnable and deeper versions of these models where we can optimize separate spatial and channel covariances of the convolutional weight distributions. We find that these networks can perform comparatively with conventional ANNs while dramatically reducing the number of trainable parameters. Channel covariances are most influential, and both weight and activation alignment are needed for classification performance. Our work outlines how neuro-inspired configurations may lead to better performance in key cases and offers a potentially tractable reduced model for ANN learning.

## 1 Introduction

Deep neural networks (DNNs) achieve state-of-the-art results, at the cost of large numbers of learned parameters and complex input-output mappings that are difficult to understand. These interpretability issues impact the trustworthiness of deep models. A key goal of neural network theory is to understand the roles of architecture and weight structure in DNNs to help bridge the gap between performance and interpretability. Among the extant theory for deep networks are kernels (Neal, 1995; Rahimi & Recht, 2007; Jacot et al., 2018) and wavelet scattering (Mallat, 2012; 2016). Kernels define a reproducing kernel Hilbert space (RKHS) for functions, which can be used to characterize the properties of neural network typically at or near their random initialization. Functions that have small norms in the RKHS are easier to learn with fewer samples and have stronger generalization abilities. Wavelet scattering is a type of deep convolutional network (CNN) that cascades fixed wavelet filters with complex modulus nonlinearities and low-pass filters to separate image variation

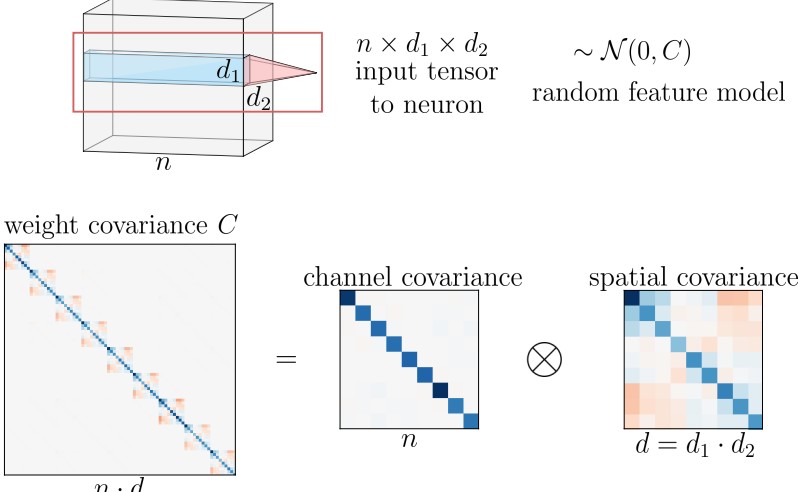

Figure 1: Learnable random feature convolutional networks: For a given layer representation, weights in our networks are modeled as drawn from correlated Gaussians where the covariance may be learned. We separably factor the weight covariance in a convolutional layer (red box) into two parts: the covariance across $n$ channels and the spatial covariance across $d = d_1 \cdot d_2$ pixels in an input patch. The spatial and channel covariances may be fixed at initialization or learned.

across scales. These networks build invariance to deformations, are mathematically tractable, and rely on fewer parameters to achieve impressive performance, particularly in low-data regimes.

Whether or not the representations in the brain are "aligned" to those in DNNs is an interesting and complicated question (Sucholutsky et al., 2023). Despite the name, neural networks on computers typically share an only superficial resemblance to the brain.

In biological neurons, receptive fields are localized areas where stimuli evoke inhibitory or excitatory neuronal responses. Visual receptive fields have been optimized over millions of years to detect small visual changes including edges, orientations, and spatial frequencies (Hubel & Wiesel, 1959). They are classically understood as comparable to wavelets, which are selective to scale and orientation (Olshausen & Field, 1996). We connect these ideas to random feature networks (RFNs), where hidden-layer weights are drawn from random distributions and only the output weights are learned. Such structured RFNs have kernels that vary depending on weight distribution and describe their function space at large width (Pandey et al., 2022).

We build randomized scattering networks (RSNs) with weight covariance modeled from biological receptive field data and identical architecture to the wavelet scattering transform. The frozen filters in these models are more interpretable than trained weights due to their wavelet eigenbasis and neural inspiration. We compare V1-like RSNs against scattering networks and uncorrelated RSNs on image classification. We find that the RSNs outperform scattering in most cases and biologically-inspired weights may improve performance.

We further investigate the importance of covariance structure by building deeper CNNs where the weight covariances are *factored* into a product of spatial and channel covariances (Fig. 1). By allowing networks to learn these covariance matrices, we gain improved performance and find that learning the channel part is more important than the spatial covariance in a deep architecture.

Finally, we try aligning resampled and trained factored networks. Guth et al. (2023) introduced "rainbow networks" that cascade random feature maps with colored weight covariance matched to the covariance structure of traditionally trained CNNs. These networks assume that the weight dependencies across layers are reduced to rotations of independent random feature matrices which align the input activations. Rainbow networks thus model the joint probability distribution of the weights of trained networks. The authors sampled new weights from the learned weight distributions and found that the sampled networks retrieve similar performance to the trained models after feature

alignment. Applying those techniques to our model, we show a collapse of this approach in later layers of the sampled networks. Alignment to the reference network's features is not sufficient for good performance in deep networks.

Our main contributions are as follows: 1) We compare the performance of RSNs with different weight distributions and traditional wavelet scattering transforms, finding similar performance across weight initializations and improvements over scattering in most cases. 2) We build general randomized CNNs with learnable covariances, separating the importance of spatial and channel covariances. These layers can be easily swapped in for convolutional layers in any architecture, and they reach comparable performance with fewer learnable parameters, e.g. 18x fewer learnable parameters leads to only 3% reduction in accuracy on our widest ResNet-18. 3) We test two major modifications to the rainbow alignment procedure which better stabilize calculations over layers and avoid the aforementioned representation collapse problem. Our central finding is that learning covariance matrices within RFNs offers a path to achieve similar accuracy to traditional ANNs while simultaneously bridging connections to theory and neuroscience.

## 2 RELATED WORK

A number of previous works have used correlated Gaussian structure in multilayer networks. Jacobsen et al. (2016) showed that learnable weights structured from a fixed Gaussian derivative basis improve performance over unstructured weights in limited dataset sizes and show improvements against wavelet scattering on large datasets. Feinman & Lake (2019) used a correlated multivariate Gaussian distribution to incorporate smooth and spatially correlated structure seen in V1 receptive fields into random convolutional weights, which improves generalization performance. Garriga-Alonso & van der Wilk (2021) found that spatial weight correlations are useful in deep networks, and Fortuin et al. (2022) show that using spatially correlated Gaussian priors in Bayesian neural networks improve performance compared to isotropic Gaussian priors.

Pandey et al. (2022) explored the use of a correlated Gaussian model to generate random receptive fields inspired by neurons in the mouse primary visual cortex (V1). Structured random feature networks have fixed hidden weights sampled from a Gaussian process with a covariance modeled from experimental receptive field data space. The weight covariances strongly impact the performance of these networks by rotating and filtering the input before random projection. In single hidden layer networks, weights with V1-inspired covariance achieve faster training times and higher accuracies than uncorrelated weights.

We also draw upon prior work related to wavelet scattering. Angles & Mallat (2018) find that embedding images using a wavelet scattering transform achieves strong performance in image generation without the use of a discriminator or encoder. Gauthier et al. (2022) show that learning the wavelet filters via parameterization can improve performance over traditional wavelet scattering filters. Li & Bonner (2022) find that the wavelet scattering transform is a strong and interpretable method for deep learning models related to V1.

## 3 METHODS

Our networks are multilayer CNNs, although nothing prevents our models from applying to fully-connected or other layers. Layers are indexed as $l = 0, \dots, L$, where $l = 0$ and $l = L$ refer to the input and output layers of the network, respectively. Convolution is assumed to operate on patches of size $d_1 \times d_2$ containing $d = d_1 \cdot d_2$ pixels per patch, $n_{l-1}$ input channels and $n_l$ output channels. Each $n_l \times n_{l-1} \times d_1 \times d_2$ weight tensor is thus equivalent to an $n_l \times (n_{l-1}d)$ matrix $W_l$. Layer $L$ is always a standard linear readout layer trained with standard methods.

### 3.1 NETWORKS WITH CORRELATED GAUSSIAN WEIGHTS

We build networks with different kinds of uncorrelated and correlated random weights. Uncorrelated weights are drawn from either Gaussian or uniform distributions. The correlated weight model draws the weights onto a neuron at layer $l$ as $w \sim \mathcal{N}(0, C_l)$, where $C_l$ is a positive semi-definite $(n_{l-1}d) \times (n_{l-1}d)$ covariance matrix (Fig. 1).

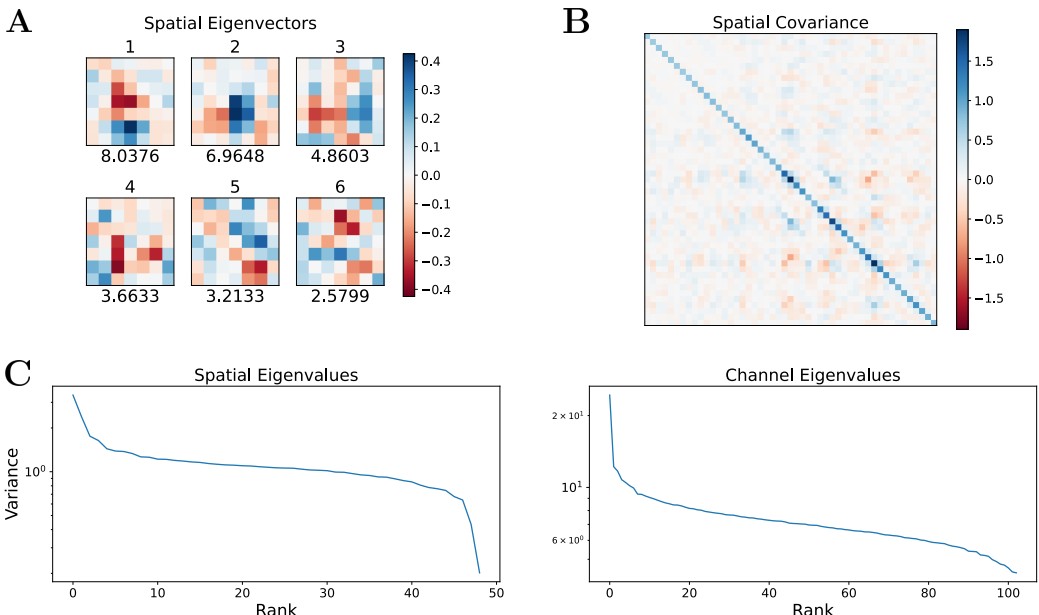

Figure 2: Spectral structure of the learned weight covariance in the convolutional layers of RSN trained on CIFAR10. (A) The top six spatial eigenvectors in the first convolutional layer, with their eigenvalues underneath. The leading eigenvectors are reminiscent of edge detecting filters. (B) The spatial covariance matrix in the first layer for a $7 \times 7$ filter. (C) The eigenvalue distributions for both the spatial and channel covariance of the second layer. Both are effectively low-dimensional.

For V1-like weights, we use a model of receptive field data from mouse V1 (Pandey et al., 2022). In this setting $C_{\text{V1}}$ depends on two parameters, the receptive field size $s$ and spatial bandwidth $f$. We ran a hyperparameter sweep over these parameters for an RSN trained on CIFAR10, finding the optimal values $s = 2$ and $f = 0.1$. We use these values across all V1-like initializations. The eigenbasis of V1-like random weights consists of Hermite wavelets.

We consider just multilayer feedforward networks. If $C_l = R_l^T R_l$ is the Cholesky square root of the covariance of the layer $l$ weights $W_l$, then the length $n_l$ feature vector $z_l$ satisfies the recursion

$$z_l = \rho(W_l z_{l-1}) = \rho(G_l R_l z_{l-1}), \tag{1}$$

where $\rho(\cdot) = \max(\cdot, 0)$. The Gaussian part $G_l$ is an $n_l \times (n_{l-1}d)$ matrix of i.i.d. entries drawn from a univariate Gaussian that is *not learned with backpropagation*. We discuss the structured convolutional RFN kernel which can be applied to our networks in Appendix A.6.

## 3.2 LEARNING THE COVARIANCE MATRIX

In an RFN, the weights at all layers except the readout are frozen. To soften this constraint, we allow the covariance $C_l$ but not the initialization $G_l$ to be learned. For CNNs, we factor $R_l$ into the separable product of two lower-dimensional matrices, an $n_{l-1} \times n_{l-1}$ channel factor $R_1$ and a $d \times d$ spatial factor $R_2$[1], as shown in Fig. 1. This yields $R_l = R_1 \otimes R_2$ and $C_l = (R_1 \otimes R_2)^T (R_1 \otimes R_2)$.

We build factored learnable covariance layers to replace traditional `Conv2d` layers in PyTorch (Paszke et al., 2019). These have $R_1$ and $R_2$ as parameters. In our parameterization, the diagonals of $R_1$ and $R_2$ are exponentiated and the lower triangular part is zero; these are initialized as the identity. The matrix $G$ is initialized with `kaiming_normal_` default scaling. We test a V1-like initialization with $R_2$ from the Cholesky decomposition of $C_{\text{V1}}$ and test networks where channel and spatial covariances ($R_1$ and $R_2$) are learned (LC and LS) or unlearnable (UC and US).

---

[1]Here, we do not explicitly show the dependence of $R_1$ and $R_2$ on the layer, although we allow this to vary.

### 3.3 RESAMPLING AND ALIGNING NETWORKS

From a multilayer network with fixed covariance structure, we create a new network where the random matrices $G'$ are resampled. Our goal is to see whether the new untrained network can recover the performance of the trained network purely through alignment, defined as follows: Given two random vectors $\phi$ and $\phi'$, we align by solving the orthogonal Procrustes problem

$$\min_{A:A^T A=I_n} \mathbb{E}\|A\phi' - \phi\|_F^2,$$

which leads to $A = EF^T := A(\phi, \phi')$ where $E$ and $F$ are the orthogonal matrices of left and right singular vectors of the cross-correlation matrix $\mathbb{E}[\phi(\phi')^T]$. If the features are deterministic matrices $\Phi$ and $\Phi'$, then using the SVD of $\Phi(\Phi')^T$ minimizes $\|A\Phi' - \Phi\|_F^2$.

We consider three different kinds of alignment: activation alignment (AA, $\phi = z$), and weight alignment on the input (IWA, $\Phi = G^T$) and output (OWA, $\Phi = G$) dimensions. Guth et al. (2023) studied AA but for reference networks without enforced Gaussian weight structure. In all cases, we align earlier before later layers and adapt batchnorm statistics in-between. The final layer weights are copied from the reference model: $W_L' = W_L$.

#### 3.3.1 ACTIVATION ALIGNMENT

In the resampled network, the weights $W_l' = G_l' R_l$ and features $z_l' = \rho(W_l' z_{l-1}')$. To align activations, we average over both batches and spatial locations when computing the cross-correlation $\mathbb{E}[z_l(z_l')^T]$ (as in Guth et al., 2023). With some abuse of notation, we will still write $A = A(z_l, z_l')$, but the $n_l \times n_l$ alignment acts on only the channel dimension as $(A \otimes I_d)z_l'$. In our code this is implemented as a forward pre hook to layer $l + 1$. Equivalently, we may absorb the alignment into $R_{l+1}$ of the next layer:

$$z_{l+1}' = \rho(W_{l+1}'(A \otimes I_d)z_l') = \rho(G_{l+1}R_{l+1}(A \otimes I_d)z_l'), \tag{2}$$

where $R_{l+1}(A \otimes I_d) = R_{1,l+1}A \otimes R_{2,l+1}$. By taking the Cholesky decomposition of $(R_{1,l+1}A)^T R_{1,l+1}A$, a new upper-triangular matrix $\hat{R}_{1,l+1}$ can be stored to make an equivalent factored covariance model.

#### 3.3.2 WEIGHT ALIGNMENT

Given a resampled matrix $G'$ and reference $G$, we also consider aligning these matrices directly. IWA does this on the input side of the weights: $\hat{G} = G'A$, with $A = A(G^T, G'^T)$, acting on the $(n_{l-1}d)$-dimensional row-space of the initialization matrices. On the other hand, OWA treats the output side of the weights: $\hat{G} = AG'$, with $A = A(G, G')$ acting on $n_l$-dimensional space. In either case, we directly replace $G'$ with $\hat{G}$ in the corresponding layer.

### 3.4 ARCHITECTURES: RANDOMIZED SCATTERING NETWORK AND RESNET

RSNs are CNNs with random feature layers in the same architecture as a wavelet scattering network. We test an RSN composed of two convolutional layers and compare these networks against 2-scale wavelet scattering networks from Kymatio (Andreux et al., 2020). We build versions of these networks (RSN-Fact) where the convolutional layers are replaced by factored covariance layers. Additional details of the architecture and parameter counts are given in Appendix A.1. To explore the effect of our structured weights on deeper networks, we also test ResNet-18 models (He et al., 2015). We replace the convolutional layers with factored covariance layers (ResNet-Fact), disabling the learning of $R_2$ in the $1 \times 1$ skip connection layers.

### 3.5 EXPERIMENTS

We run image classification experiments on the CIFAR10 and CIFAR100 datasets (Krizhevsky, 2009). In RFNs, all weights except the readout are kept at their initialization. In all other models except traditional networks, the weight initializations $G_l$ are fixed while the $R_l$ are learned. To test performance in the limited data regime, we evaluate our models on CIFAR10 with small sample

size (CIFAR10-S) using only 50 training samples per class. To understand variance due to initialization and training process, we train ten different networks on each dataset and report the average and standard error accuracy across seeds. We run experiments with RSN models initialized with V1-like weights, Gaussian weights, and uniform weights, and a two-scale wavelet scattering network. We include results of our RSN-Fact and ResNet-Fact models with V1-like spatial covariance initialization.

Additionally, we run unsupervised learning experiments comparing RSNs to the wavelet scattering embedding described in Angles & Mallat (2018) and see similar results across image generation tasks (Fig. 9 in Appendix A.5).

We also run extensive resampling and alignment experiments for our various models and alignment methods (AA, IWA, OWA). After alignment, we then fine-tune the readout layer. We estimate a sampled accuracy and deviation using 5 different samples of models generated from this process, before finetuning the readout layer over ten epochs of the training set.

The code for our experiments and to implement these networks and learning rules is available at `https://github.com/glomerulus-lab/fact-conv/`.

## 4 RESULTS

### 4.1 LEARNING COVARIANCE STRUCTURE FOR IMAGE CLASSIFICATION

We train RSN, RSN-Trad, RSN-Fact, ResNet-Trad, and ResNet-Fact models on CIFAR10, CIFAR10-S, and CIFAR100. Visualizations of the eigenspectra for the learned spatial and channel covariance matrices in an RSN-Fact network are shown in Fig. 2 (see also Fig. 7 in Appendix A.2). The leading spatial eigenvectors are relatively smooth and display characteristics of edge and blob detectors. The spectra of our spatial and channel matrices are effectively low-dimensional, with a few leading components accounting for most of the variance. These results are consistent across layers and initializations. Although the exact structure of the eigenvectors varies, similar patterns emerge across initializations (not shown).

Image classification results are displayed in Fig. 3 for models with fixed features (left) and learnable features (right), including RSN, wavelet scattering, and ResNet architectures. Among the fixed-feature RSN models with different weight distributions, we see moderate improvement with V1-like initialization on CIFAR10. All of the RSN models outperform scattering except on CIFAR10-S, the small data setting where scattering is known to be powerful (Oyallon et al., 2019; Gauthier et al., 2022).

When features are learnable and we use lots of data (all settings except CIFAR10-S), all variations of the factored covariance models outperform the fixed-feature networks. More learning tends to improve performance for both RSN and ResNet. On the other hand, when in the small sample setting, bigger models and more learning do not lead to good performance due to overfitting. ResNet-Fact models performs worse than RSN-Fact on CIFAR10-S. RSN-USLC outperforms the other learnable models, but it is still a 3% worse than wavelet scattering. This hints that the other models may be overfitting spatial features. In the ResNet-Fact models, we see similar accuracies between the USLC and LSLC models among all three datasets, showing that the majority of the ResNet performance comes from learning the channel covariances.

Furthermore, in CIFAR10 and CIFAR100, these models approach (but fall slightly short of) the performance of a traditional model with 17x fewer learned parameters. Full parameter counts are shown in Table 2 in Appendix A.1.

Lastly, we compare ResNet-Fact models initialized with V1-inspired against ResNet-Fact models initialized with identity spatial covariance in Table 3 in Appendix A.3 (RSN are also shown). In this architecture, V1-like spatial covariance has only minor effects on performance.

### 4.2 RESAMPLING AND ALIGNMENT TO APPROXIMATE A REFERENCE NETWORK

We now look at the performance of resampled and aligned networks using the methods of activation alignment (AA) and input weight alignment (IWA) on ResNet-Fact models. We do not show

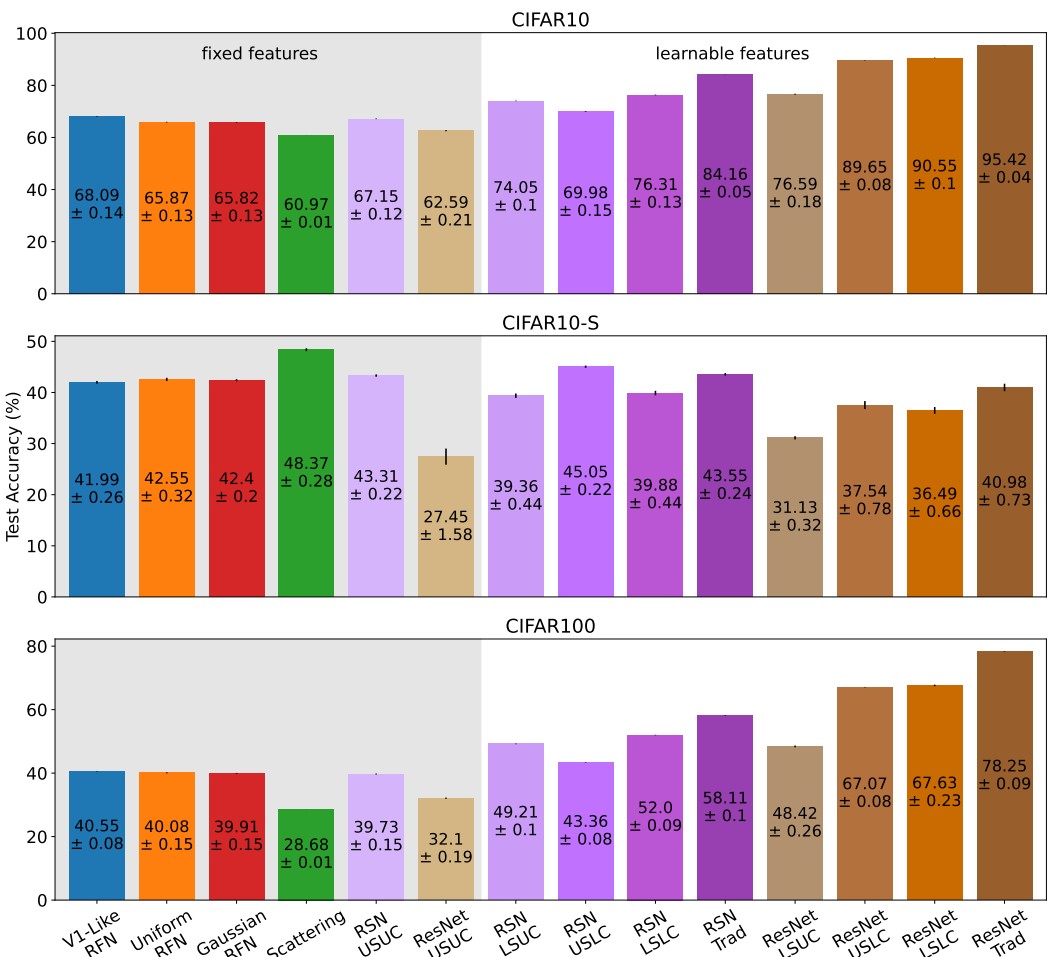

Figure 3: Classification performance of our various models—RSN with varied fixed initializations (labeled "RFN"), comparable scattering networks, RSN-Fact and ResNet-Fact models with unlearnable/learnable covariance, and their traditionally-trained counterparts ("Trad")—on test datasets. USUC = fixed spatial and channel covariances, USLC = fixed spatial and learnable channel covariances, LSUC = learnable spatial and fixed channel covariances, LSLC = learnable spatial and channel covariances. Fixed and learnable features maps are shown at left and right. Test accuracies are shown averaged over ten trials with standard error. Learnable features and more complex models tend to help except in the small sample setting. Our ResNet-Fact models are more lightweight than traditional models: Resnet LSLC achieves within 5% accuracy with 17x fewer trainable parameters.

OWA, since it never achieved high accuracy in any of our experiments. Fig. 4, left, shows the results of applying our sampling procedure to a reference ResNet-Fact over several widths as well as ResNet-Trad for comparison. Accuracy increases with width for ResNet-Trad and ResNet-Fact. The various alignment methods show different behavior: AA never exceeds 25% accuracy, and IWA hovers between 40 and 60%, whereas IWA + AA better approximates the accuracies of the reference ResNet-Fact, reaching nearly 90% at maximum width. For the successful IWA + AA method, finetuning the linear classifier layer does not significantly change the accuracy compared to reusing the readout weights of the reference. Unsurprisingly, even the best alignment method cannot attain the full accuracy of the reference ResNet-Fact.

To understand why different alignment methods behave so differently, we inspect the intermediate representations of the resampled width-1 networks via a linear probing (Fig. 4, right). In early layers of the ResNet, probe accuracy increases as we move deeper into the network, reaching a plateau around layer 3 although at a lower accuracy for AA. Around layer 8, IWA + AA starts to

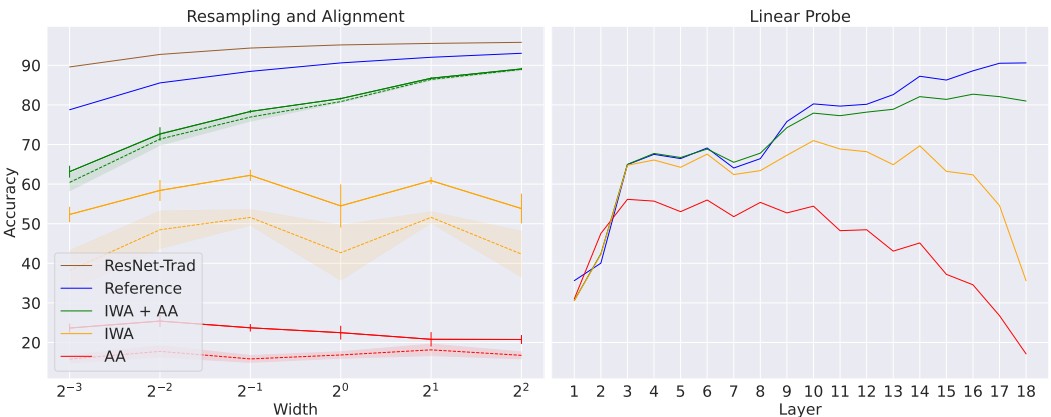

Figure 4: **Left:** Performance of Pretrained ResNets (brown), ResNet-Fact (blue) and its sampled counterparts over width. Solid/dashed lines indicate with/without fine-tuning of the readout. Error bars/color-fill indicate standard deviation of performance with/without finetuning of the the readout. Results are calculated over 5 samplings. IWA + AA does the best in approximating performance of the original network, requiring little finetuning of the linear classifier at large widths. **Right:** Linear probe performance—training the readout on intermediate layers—of various methods of resampling and aligning a pretrained network. AA reaches a low plateau and degrades roughly half-way through the layers. IWA reaches better performance and delays this degradation until later layers. IWA + AA best approximates the performance of the reference ResNet (blue).

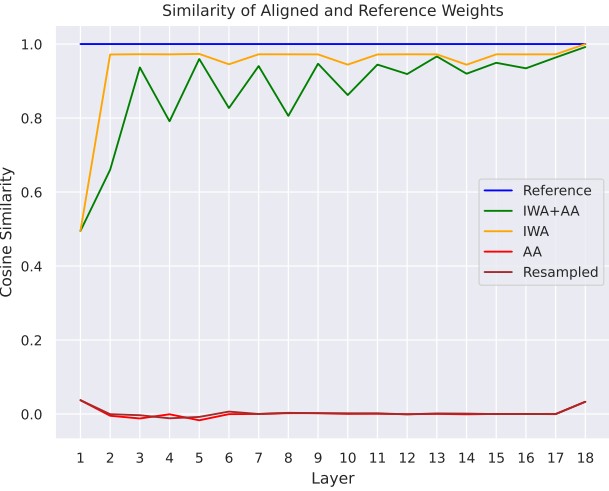

Figure 5: Cosine similarity between aligned and reference weights by layer. AA and an unaligned resampled network give weights which are nearly orthogonal to the reference width-1 ResNet. IWA and IWA + AA produces weights which nearly line up with the originator pretrained network.

beat the performance of IWA alone. Probe accuracies of AA and IWA degrade past layers 9 and 14, respectively.

Finally, Fig. 5 shows the cosine similarity of the aligned and reference weights for a width-1 ResNet across layers. The different alignment methods lead to different aligned weights $\hat{W}_l = \hat{G}_l \hat{R}_l$ that may be compared to the reference model weights $W_l = G_l R_l$; studying the composite weights captures both weight and activation alignment effects. AA fails to align the weight matrices any more than a randomly resampled network without any alignment step: aligned and reference weights are nearly orthogonal for all layers. This is expected since the first layer weights are different. IWA, on the other hand, leads to high similarity with the reference model. Interestingly, IWA + AA has

lower similarity than IWA alone but performs better in accuracy. OWA produces an intermediate value of similarity (not shown).

## 5 DISCUSSION

We use correlated Gaussian weights as a flexible model for multilayer networks where the weights are frozen or learned. This formulation is tested in Randomized Scattering Networks (RSNs) and ResNets. RSNs follow a similar architecture to the wavelet scattering transform and use fixed neuro-inspired filters as feature extractors. We find that RSNs can outperform wavelet scattering models and that biologically-inspired weights can lead to an advantage in supervised learning. To highlight the similarity between RSNs and wavelet scattering, we build image generation embeddings described in Angles & Mallat (2018), seeing similar performance with both models (Appendix A.5).

Our main finding is that introducing covariance learnability into RSN and ResNet models leads to improved performance. In particular, learning the channel covariance reaches within 5% of the performance of a traditional CNN of the same architecture. This is achieved with 17x fewer parameters than fully trained networks. In limited experiments, we find effectively low rank spatial and diagonal channel covariances, suggesting further parameter reductions to try.

We experiment with activation alignment (AA), based on pre-activations to a layer, and input weight alignment (IWA), where we directly align the weight matrices. With both of these methods, our linear probe study shows a degradation of accuracy in later layers of ResNet-Fact. Combining these two methods of alignment avoids the degradation and comes close to approximating original performance at the cost of requiring direct access to the pretrained weights and destroying their randomness. With this alignment technique, sampled weight performance converges to finetuned performance at higher width, suggesting that finetuning the classifier head isn't necessary.

Guth et al. (2023) studied AA applied to wide, moderately deep networks, where colored Gaussian models were fit to each layer of a traditionally trained network. Among other results, they found that AA alone can lead to good performance in 7-layer scattering networks. An important conclusion was that this would not affect the network kernel, since the weights at a given layer were independent once conditioned on the previous layer weights. Weight alignment breaks this kernel equivalence. There are many possible reasons for our contradictory results: narrower layers, different architecture, or the covariance factorization. However, supplemental experiments (Fig. 8) with rainbow sampling reveal similar discrepancies in ResNets. Probing readout accuracies from intermediate layers leads us to hypothesize that in deeper networks, more than AA is needed to achieve strong performance.

Limitations to our work include the few architectures that were considered and emphasis on thumbnail image datasets. We only considered the ReLU nonlinearity; other choices could lead to other results. In particular, it is unclear how the signal processing ideas underlying our work apply to very different domains such as text processing.

In future work, we plan to investigate further ways to compress networks using weight factorization. Further speedups may be possible using methods such as fast random projections (Le et al., 2013). We also want to study the randomized scattering kernel, to strengthen our theoretical understanding, and more thoroughly analyze the optimal representations and dynamics of covariance learning.

ACKNOWLEDGMENTS

V.W. was supported by a fellowship from the International Network for Bio-Inspired Computing (NSF AccelNet award 2019976) and travel grants from the WWU CS department and ASWWU Student Enhancement Fund. G.W. acknowledges support from the Canada-CIFAR AI Chair program, an NSERC Discovery Grant (RGPIN-03267), an NIH grant (R01GM135929), an NSF DMS grant (2327211), and an FRQNT NOVA grant (2023-NOVA-329125). G.L. acknowledges support from the Canada-CIFAR AI Chair program, the Canada Research Chair in Neural Computation and Interfacing, and an NSERC Discovery Grant (RGPIN-2018-04821). We thank Mick Bonner, Florentin Guth, Brice Ménard, and Mark van der Wilk for discussions. Our code for rainbow sampling was based off the CCN tutorial on the Bonner lab website. Thank you to Olexa Bilaniuk for support with the Mila cluster.

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

Jacob A Zavatone-Veth and Cengiz Pehlevan. Learning curves for deep structured gaussian feature models.

## A APPENDIX

### A.1 NETWORK ARCHITECTURES

The RSN models are composed of an input layer followed by two convolutional layers with $(7 \times 7)$ kernels, a hidden dimension of 100, a stride of 1, and padding set to 3. Each of these layers go through batch normalization and average pooling before being flattened, concatenated, and passed through a linear classifier. Fig. 6 models the architecture, which is similar to the scattering transform with scale $J = 2$ and input passed through convolution, nonlinearity, and averaging. We train all RSNs for 90 epochs using a stochastic gradient descent optimizer and update the learning rate by 0.2 every 20 epochs. Table 1 displays the number of learnable parameters in the feature and classifier layers of the RSN along with a two-scale wavelet scattering model, RSN-Fact models with learnable and unlearnable spatial and channel covariances, and a traditional version of RSN with learnable weights in the convolution layers.

The ResNet models follow the architecture in He et al. (2015). We train all ResNets for 200 epochs using stochastic gradient descent and update the learning rate using a cosine annealing scheduler. We explore the effect of width by applying a scale factor to the number of channels in the convolutional models. Table 2 displays the number of learnable parameters in variations of ResNet-Fact across widths ranging from $2^{-3}$ to $2^2$. Notably, we see a 14-18x decrease in learnable parameters between LSLC and a fully trained ResNet, with the compression factor increasing with width.

### A.2 LEARNED SPATIAL EIGENVECTORS

Fig. 7 displays the top six learned spatial eigenvectors in the second RSN convolutional layer (left) and the first ResNet convolutional layer (right). We observe less smoothness than in the first RSN layer (Fig. 2). The ResNet spatial filters are $3 \times 3$ so these eigenvectors are of a smaller shape, but the first could be interpreted as a centered blob detector. We compared eigenvectors across different random initializations, noting similar patterns across layers (not shown).

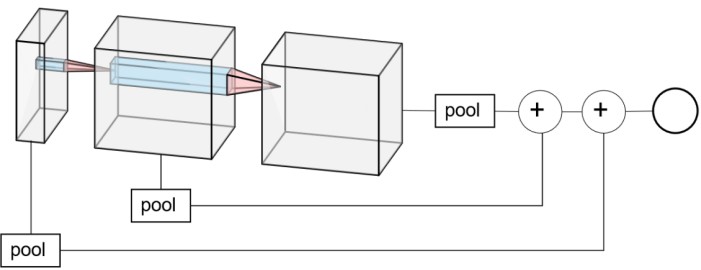

Figure 6: Network diagram for RSN. Two convolutional layers are fed through BatchNorm (not shown), then their outputs and the input are pooled and concatenated before passing to the classifier. Note that the "+" symbols correpond to concatenation, not summation as in a ResNet.

|  |  | RSN | Scattering | USUC | LSUC | USLC | LSLC | Trad |
|---|---|---|---|---|---|---|---|---|
| CIFAR10 | Feature | 618 | 0 | 618 | 3.07k | 5.98k | 8.43k | 520k |
|  | Readout | 130k | 32.7k | 130k | 130k | 130k | 130k | 130k |
| CIFAR100 | Feature | 618 | 0 | 618 | 3.07k | 5.98k | 8.43k | 520k |
|  | Readout | 1.30M | 327k | 1.30M | 1.30M | 1.30M | 1.30M | 1.30M |

Table 1: The number of learnable parameters in the feature and readout layers of each of our 3-layer RSN models for CIFAR10 and CIFAR100 plus the single-scale wavelet scattering model. In the fixed features and USUC columns, the learnable feature channels are due to batch normalization layers. The rightmost column shows the number of learnable parameters in a fully-trained RSN.

| Width | USUC | LSUC | USLC | LSLC | Trad | Compression |
|-------|------|------|------|------|------|-------------|
| $2^{-3}$ | 1.85k | 2.62k | 11.6k | 12.4k | 176k | 14.2 |
| $2^{-2}$ | 3.69k | 4.46k | 42.3k | 43.1k | 701k | 16.3 |
| $2^{-1}$ | 7.37k | 8.14k | 161k | 162k | 2.80M | 17.3 |
| $2^{0}$ | 14.7k | 15.5k | 627k | 628k | 11.2M | 17.8 |
| $2^{1}$ | 29.5k | 30.2k | 2.48M | 2.48M | 44.7M | 18.0 |
| $2^{2}$ | 58.9k | 59.7k | 9.83M | 9.83M | 178M | 18.1 |

Table 2: The number of learnable parameters in ResNet-Fact models trained on CIFAR10 at different widths compared to traditional models. The compression ratio of LSLC learnable parameters relative to traditional increases with width; this ratio is higher for USUC, LSUC, and USLC models.

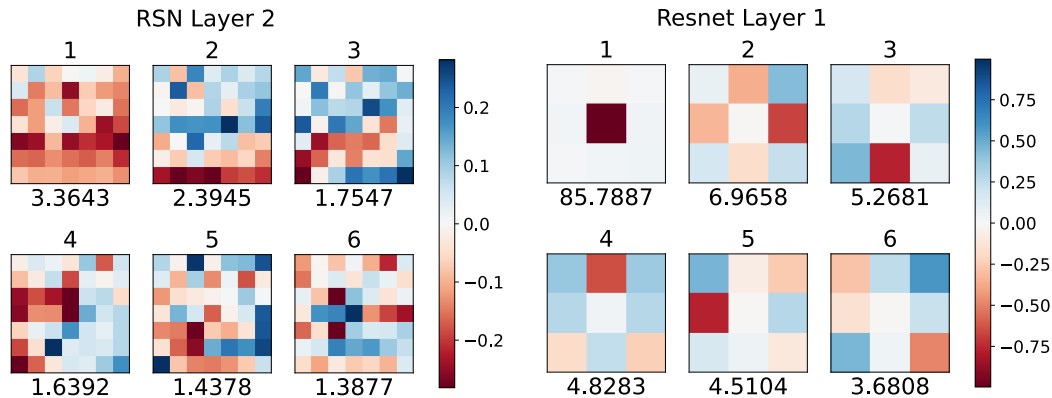

Figure 7: Leading learned spatial eigenvectors. **Left:** The second RSN-Fact convolutional layer. We see less smoothness than the first-layer eigenvectors (Fig. 2). **Right:** The first layer of a ResNet-Fact. The ResNet is dominated by a 1x1 centered bump eigenfeature with significantly outlying eigenvalue.

### A.3 V1-LIKE VS DEFAULT INITIALIZATION

Table 3 shows the result of default versus V1-like initialization with ResNet-Fact and RSN-Fact models for the spatial covariance matrix $R_2$. We see that the V1-like initialization may lead to slightly improved performance. In the ResNet models across both initializations, we see close performance between USLC and LSLC, indicating that learning channels is beneficial. However, in most of the RSN models (CIFAR10 and CIFAR100), we see that the LSUC models reach the closest performance to LSLC, indicating that spatial covariance may be more important in this architecture.

### A.4 NETWORK RESAMPLING

Fig. 8 shows a supplemental experiment applying the method of Guth et al. (2023) to ResNet-18. In this scenario, we trained a ResNet-Trad model then at each layer fit a Gaussian covariance to the trained weights and resample new weights accordingly. After weight sampling, we apply the same alignment methods as studied in the main text. Our code was adapted from that provided at `https://bonnerlab.github.io/ccn-tutorial/pages/analyzing_neural_networks.html`.

Each linear probe (Alain & Bengio, 2018; Richter et al., 2021) trains read outs from the feature map of some convolutional layer. For simplicity, we don't show the skip-connection layers in ResNet models. All linear probes are trained over 200 epochs with learning rate $10^{-4}$. The model parameters and buffers are held fixed as it would be on evaluation.

|          |                      | USUC             | LSUC             | USLC             | LSLC             |
| -------- | -------------------- | ---------------- | ---------------- | ---------------- | ---------------- |
| CIFAR10  | ResNet-Fact V1       | $62.59 \pm 0.21$ | $76.59 \pm 0.18$ | $89.65 \pm 0.08$ | $90.55 \pm 0.10$ |
|          | ResNet-Fact Default  | $62.50 \pm 0.34$ | $76.36 \pm 0.11$ | $89.52 \pm 0.10$ | $90.59 \pm 0.08$ |
|          | RSN V1               | $67.15 \pm 0.12$ | $74.05 \pm 0.1$  | $69.98 \pm 0.15$ | $76.31 \pm 0.13$ |
|          | RSN Default          | $66.03 \pm 0.07$ | $71.86 \pm 0.07$ | $68.6 \pm 0.08$  | $74.0 \pm 0.11$  |
| CIFAR SS | ResNet-Fact V1       | $27.45 \pm 1.58$ | $31.13 \pm 0.32$ | $37.54 \pm 0.78$ | $36.49 \pm 0.66$ |
|          | ResNet-Fact Default  | $25.01 \pm 1.63$ | $29.55 \pm 1.01$ | $37.05 \pm 0.75$ | $35.97 \pm 0.65$ |
|          | RSN V1               | $43.31 \pm 0.22$ | $39.36 \pm 0.44$ | $45.05 \pm 0.22$ | $39.88 \pm 0.44$ |
|          | RSN Default          | $42.66 \pm 0.32$ | $44.44 \pm 0.31$ | $44.1 \pm 0.32$  | $43.4 \pm 0.55$  |
| CIFAR100 | ResNet-Fact V1       | $32.10 \pm 0.19$ | $48.42 \pm 0.26$ | $67.07 \pm 0.08$ | $67.63 \pm 0.23$ |
|          | ResNet-Fact Default  | $32.49 \pm 0.21$ | $47.66 \pm 0.24$ | $66.47 \pm 0.08$ | $67.99 \pm 0.13$ |
|          | RSN V1 Default       | $39.73 \pm 0.15$ | $49.21 \pm 0.1$  | $43.36 \pm 0.08$ | $52.0 \pm 0.09$  |
|          | RSN Default          | $38.76 \pm 0.09$ | $46.09 \pm 0.12$ | $42.23 \pm 0.09$ | $49.22 \pm 0.1$  |

Table 3: We test the performance of V1-like spatial covariance initialization in the $R_2$ matrix versus the default covariance initialization on models with factored learnable covariance layers. Accuracy is averaged over ten seeds with standard error reported. V1-initialization may improve performance slightly, but its effects are weak especially in the ResNet.

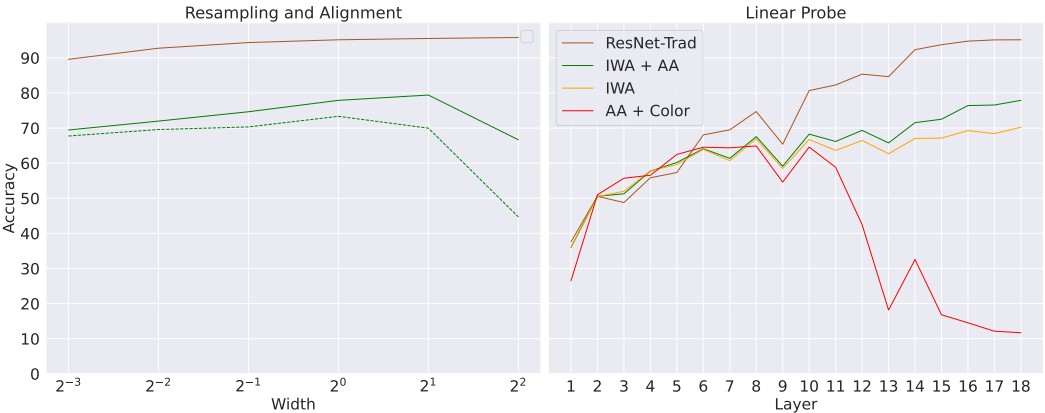

Figure 8: The rainbow sampling method of Guth et al. (2023) applied to ResNet-18. **Left:** The accuracy of traditional ResNets and their resampled and aligned (dashed) and resampled, aligned, and fine-tuned (solid) counter parts over width. At larger widths we are unable to approximate the performance of larger width networks. **Right:** We inspect the linear probe performance on various methods of sampling a network from a pretrained convolutional network. AA accuracy declines after layer 11. Aligning a sampled weight with a corresponding pretrained network's layer weight, IWA, prevents this decline. IWA + AA gives closest performance to that of the originator pretrained network, but a large difference remains in later layers.

## A.5 UNSUPERVISED LEARNING

To test our RSNs against wavelet scattering in the unsupervised learning domain, we build a scattering-based image generator described in Angles & Mallat (2018). Their network provides a strong mathematical framework with similar properties to generative adversarial networks (GANs) and variational auto-encoders (VAEs). In their model, images are embedded with a wavelet scattering transform, then passed through an affine whitening layer. The whitened scattering representation is then inverted by the generator model, which is composed of 5 convolutional layers with upsampling, reflection padding, batch normalization, and ReLU nonlinearity.

We investigate whether embedding the input images with fixed V1-like features achieves similar performance to the wavelet scattering embedding in three image generation areas: linear interpolation between two images, generating new images from Gaussian white noise, and reconstructing images from the training and test sets. We use the wavelet scattering package Kymatio (Andreux et al., 2020) to regenerate results from Angles & Mallat (2018). In our RSN version of the wavelet scat-

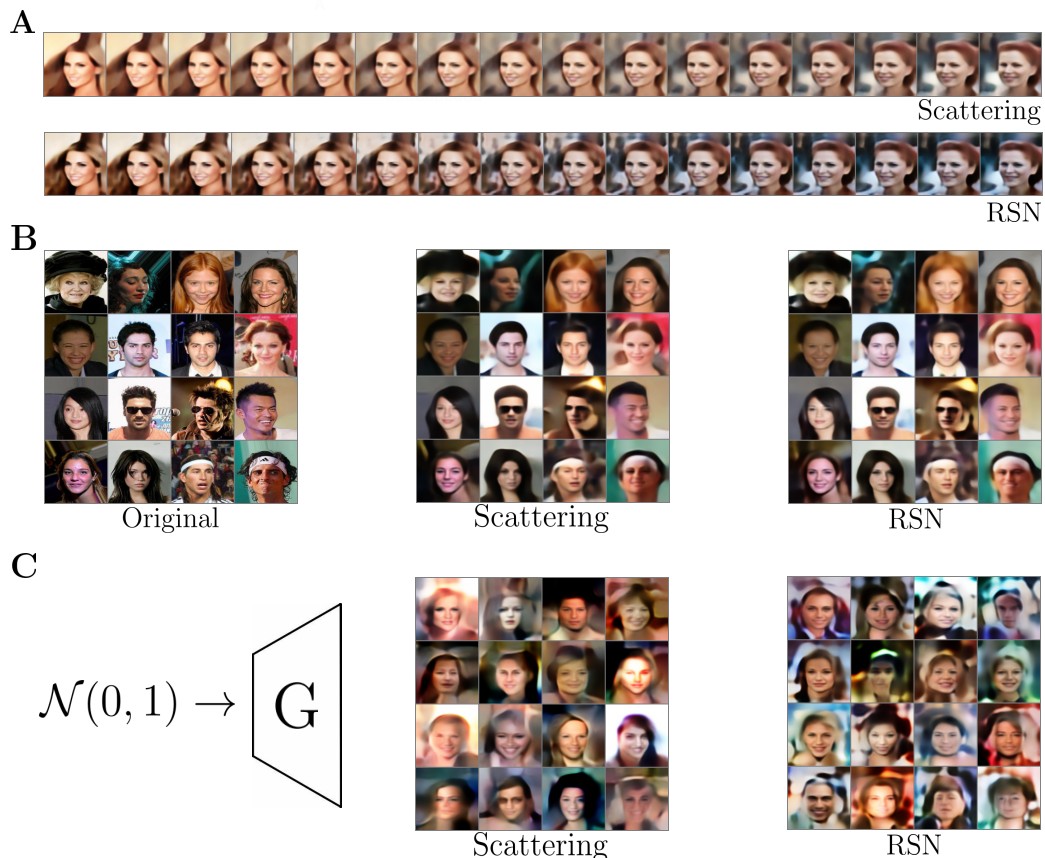

Figure 9: Autoencoding with either wavelet scattering or randomized scattering networks produces similar results. (A) Linear interpolation between two training images. (B) Reconstructing a set of images from the test set. (C) New images generated from white noise. Left, the results of scattering embedding and right, the results of our V1-like RSN embedding. The V1-like generated images are less realistic.

tering embedding, the input image is passed through two convolutional layers with frozen V1-like weights. We implement whitening using `IncrementalPCA` from scikit-learn (Pedregosa et al., 2011). We test our image generation model on the CelebA dataset (Liu et al., 2015), composed of over 200,000 images of celebrity faces.

Visually, we see similar results between our generated results and the wavelet scattering results (Fig. 9), showing that fixed V1-like features work well as an embedding similar to scattering. However, our train PSNR score on CelebA is 30.45 and our test score is 28.89, while the scores in our reconstructed network from Angles & Mallat (2018) are 30.21 for train and 29.91 for test, so our networks perform slightly worse in this regard.

## A.6 KERNELS OF MULTILAYER STRUCTURED RANDOM FEATURE NETWORKS

Kernels, e.g. the original neural network Gaussian process (NNGP, Neal, 1995) or the neural tangent kernel (NTK, Jacot et al., 2018), are a popular way to analyze function spaces associated with neural network architectures. The layer-$l$ NNGP is given by the infinite width limit $n_l \to \infty$ of the scaled inner product of hidden layer features $z_l$ for two different inputs $z_0 = x$ and $z_0 = x'$:

$$k_l(x, x') = \frac{1}{n_l}\langle z_l(x), z_l(x')\rangle \approx \mathbb{E}_{W_1 \dots W_l}[z_l(x)z_l(x')], \qquad (3)$$

which in a feedforward network depends on the the previous and current layer weights. In certain settings NNGP and NTK kernels can be expressed recursively. With correlated Gaussian features,

this can be evaluated as the kernel for unstructured features acting on inputs $z_{l-1}(x)$ transformed as $R_l z_{l-1}(x)$ (Pandey et al., 2022).

For a network with convolutional structure, $z_l[p] = \rho(G_l \Lambda U z_{l-1}[p])$ where $p$ is a patch of the representation at each layer. This implies a convolutional kernel representation (Mairal et al., 2014; Guth et al., 2023):

$$k_l(x, x') = \sum_p \mathbb{E}_{g \sim \mathcal{N}(0,1)}[\rho(\langle R_l z_{l-1}(x)[p], g \rangle)\rho(\langle R_l z_{l-1}(x')[p], g \rangle)]. \tag{4}$$

Note that the structure of this function is additive (Harris, 2019) over patches. Concatenating features, as in our RSN, yields a kernel of the form:

$$k(x, x') = x^T x' + \sum_{l=1}^{L} k_l(x, x'). \tag{5}$$

An interesting direction for future work is to study how the transformation induced by the covariance of the correlated weights ($R_l$ or equivalently the spectral decomposition of the covariance $C_l$) affects the RKHS associated to the network. Perhaps the techniques of Xiao (2021), who studied CNNs without covariance structure, are applicable. Additionally, we could apply methods from Bosch et al., Schröder et al., and Zavatone-Veth & Pehlevan, who computed sharp asymptotics for the generalization performance of networks with random weights. The former two works consider nonlinear models with i.i.d. weights, while the latter considers correlated Gaussian weights.

