# OpenReview forum: "Learning and Aligning Structured Random Feature Networks"
_ICLR.cc/2024/Workshop/Re-Align — ICLR 2024 Workshop Re-Align Poster_

### Official Review · Reviewer_vQWP · 2024-02-21
**Review of "Learning and Aligning Structured Random Feature Networks"**

**Rating:** 3
**Fit:** 3
**Confidence:** 3

**Workshop Review:**

This paper proposes a class of wavelet scattering networks with structured covariances. This paper is clearly relevant and interesting to the audience of this workshop, as it combines features of models commonly used in neuroscience and in machine learning (I have in mind shallower LN models and rainbow networks, respectively). The experiments are clearly described and appear well-motivated.

I have only a few, relatively minor comments:

- In discussing related works, it would be useful to mention the line of work in theoretical machine learning on computing sharp asymptotics for the generalization performance of networks with random weights. I have in mind Schröder et al. ICML 2023, Bosch et al. 2023 (on arXiv), and Zavatone-Veth & Pehlevan NeurIPS 2023. The former two works consider nonlinear models with i.i.d. weights, while the later considers correlated Gaussian weights. Given appropriate Gaussian equivalence results, one could compute the asymptotic generalization error for one of the models considered here given knowledge of the adapted covariance spectrum.

- In Figure 3, it might be clearer to use a shaded background to highlight which models have fixed feature maps.

- Do you understand why the alignment for IWA+AA in Figure 5 fluctuates from layer to layer in such a pronounced way? Also, the label hear reads AA+IWA rather than IWA+AA.

- How sensitive are the results presented to the choice of ReLU nonlinearity?

- For a broad audience, it would be useful to discuss rainbow networks in greater depth.

**Reason For Not Giving Higher Score:**

N/A

**Reason For Not Giving Lower Score:**

The paper is reasonably clearly written, and is relevant to the workshop audience.

**Reviewer Domain:**

neuroscience

---

### Official Review · Reviewer_WS1S · 2024-02-21
**Does not seem like a good fit to the goals of this workshop**

**Rating:** 1
**Fit:** 1
**Confidence:** 1

**Workshop Review:**

This paper investigates how neuron-inspired deep network architectures might lead to better performance. I have two main concerns. First, this does not really seem relevant to the goals of the workshop (as I understand them). Second, the empirical performance of these methods (see Fig. 3) seems very underwhelming to me. Conventional ANNs with learned features perform considerably better than the proposed method. Moreover, the authors only use CIFAR-10 and CIFAR-100 benchmarks. Comparing performance on a more extensive benchmark (e.g. ImageNet) is necessary. It is also not clear to me that the V1-like random feature model performs better than other random feature models. Indeed if I understand it correctly, the V1-like random feature model was optimized based on an experiment in mice---a species with much lower visual acuity than, for example, nonhuman primates.

I am marking this review with a low confidence score since my primary interest is not in machine learning benchmarks.

**Reason For Not Giving Higher Score:**

My main concern is the motivation of the work, which does not involve "alignment" of neural representations. I have marked this review with low confidence. If other reviewers dissent with my perspective, I'm happy to defer to them.

**Reason For Not Giving Lower Score:**

N/A

**Reviewer Domain:**

neuroscience

---

### Author Response · Authors · 2024-05-03
**Revised version**

Thank you to both reviewers for your time looking at our work and for your helpful comments. We have revised the paper to hopefully address the main issues.

* In discussing related works, it would be useful to mention the line of work in theoretical machine learning on computing sharp asymptotics for the generalization performance of networks with random weights. I have in mind Schröder et al. ICML 2023, Bosch et al. 2023 (on arXiv), and Zavatone-Veth & Pehlevan NeurIPS 2023. The former two works consider nonlinear models with i.i.d. weights, while the later considers correlated Gaussian weights. Given appropriate Gaussian equivalence results, one could compute the asymptotic generalization error for one of the models considered here given knowledge of the adapted covariance spectrum.

We have added to end (appendix): “Additionally, we could apply methods from Bosch et al., Schr ̈oder et al., and Zavatone-Veth & Pehlevan, who computed sharp asymptotics for the generalization performance of networks with random weights. The former two works consider nonlinear models with i.i.d. weights, while the latter considers correlated Gaussian weights.”
* In Figure 3, it might be clearer to use a shaded background to highlight which models have fixed feature maps.

Done
* Do you understand why the alignment for IWA+AA in Figure 5 fluctuates from layer to layer in such a pronounced way? Also, the label hear reads AA+IWA rather than IWA+AA.

We do not have a good explanation for the fluctuations from layer-to-layer. The label has been fixed, thanks.

* How sensitive are the results presented to the choice of ReLU nonlinearity?

We added this caveat to the limitations: “We only considered the ReLU nonlinearity; other choices could lead to other results.”
However, we would expect that the learnable covariance model would apply to other types of nonlinearities.

* For a broad audience, it would be useful to discuss rainbow networks in greater depth.

We added these sentences to the discussion of rainbow networks in the introduction: “These networks assume that the weight dependencies across layers are reduced to rotations of independent random feature matrices which align the input activations. Rainbow networks thus model the joint probability distribution of the weights of trained networks.”

* ImageNet?

Running on more diverse image datasets is left for future work.

-----------------
A minor error was caught in revisions: In our original paper, the ResNet-trad results in Fig 3 were a few percent lower than in the final version due to a mistake in the initialization. This has been fixed for the final version. We have also added a link to our code repository, although this remains in-progress.

---

### Decision · Program_Chairs · 2024-03-02

Accept (Poster)